# 🧠 AWARe: Mitigating Catastrophic Forgetting via Activation-Weighted Adaptive REtaining

## Abstract

Multimodal Large Language Models (MLLMs) exhibit strong generalization and reasoning abilities due to large-scale multimodal pre-training. However, fine-tuning these models on downstream tasks often leads to catastrophic forgetting, where newly learned task-specific knowledge degrades previously acquired capabilities. This issue arises because gradient updates for new tasks overwrite parameters critical to prior knowledge, limiting the practical deployment of MLLMs. To address this challenge, we propose **Activation-Weighted Adaptive REtaining (AWARe)**, a fine-tuning method that mitigates catastrophic forgetting by dynamically controlling parameter updates based on activation patterns. AWARe assigns activation-based importance scores to parameters, selectively freezing those essential for preserving prior capabilities while allowing less important parameters to adapt to new tasks. Importantly, AWARe operates *without modifying model architectures*, ensuring compatibility with existing inference engines. Extensive experiments demonstrate that AWARe effectively preserves upstream capabilities while achieving superior downstream performance compared to existing methods. Code is available at https://anonymous.4open.science/r/AWARe-FEE2/.

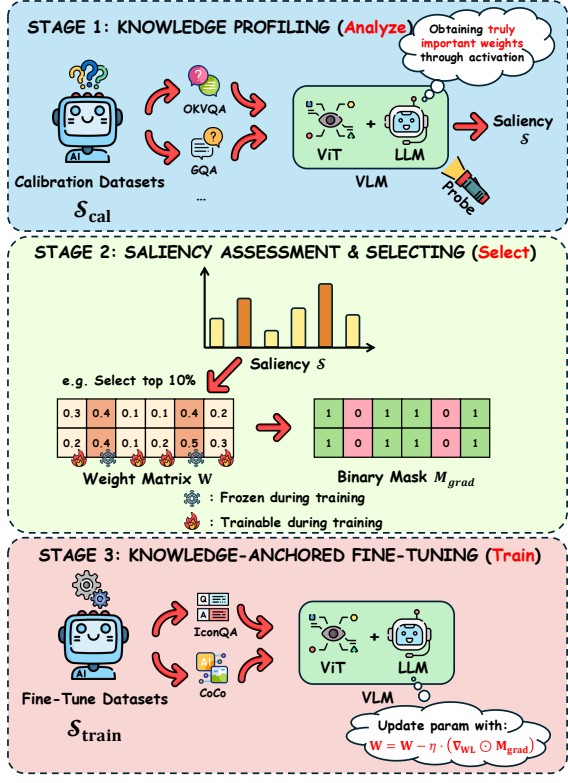

*Figure 1.* Overview of our proposed Activation-Weighted Adaptive REtaining (AWARe) framework. AWARe estimates the saliency of parameters based on task-induced activation distributions from upstream tasks. By selectively freezing highly salient parameters that capture core upstream features, the model mitigates catastrophic forgetting while remaining plastic for downstream task adaptation.

## 1. Introduction

Multimodal Large Language Models (MLLMs) have emerged as powerful versatile agents, capable of solving complex tasks that require understanding and reasoning across both inputs of vision and language (Zhu et al., 2023; Dai et al., 2023; Bai et al., 2023; Liu et al., 2023; Lin et al.,

[1]Anonymous Institution, Anonymous City, Anonymous Region, Anonymous Country. Correspondence to: Anonymous Author <anon.email@domain.com>.

Preliminary work. Under review by the International Conference on Machine Learning (ICML). Do not distribute.

2023a; Chen et al., 2024; Wang et al., 2024; Zhu et al., 2025a; Wang et al., 2025; Bai et al., 2025b;a). Typical MLLMs are composed of a pre-trained visual encoder, a large language model (LLM), and a connector module (Li et al., 2022; Dai et al., 2023; Liu et al., 2023). These models facilitate strong generalization abilities acquired from large-scale pre-training. Despite their impressive zero-shot capabilities, fine-tuning MLLMs on downstream tasks remains a standard practice to tailor models for specific domains or improve instruction-following performance (Dai et al., 2023; Bai et al., 2023; Liu et al., 2023). However,

this adaptation process often comes at a cost: *catastrophic forgetting* (CF), where the model's proficiency in previously learned upstream tasks degrades significantly as it optimizes for new objectives (McCloskey & Cohen, 1989; McClelland et al., 1995; Kirkpatrick et al., 2017; Li et al., 2024b). It is important to clarify that our focus is on *stabilization during specialization*—preserving the broad, pre-trained capabilities of an MLLM while fine-tuning on a single downstream task. This scenario is distinct from sequential continual learning, where a model must learn a series of new tasks over time. Nonetheless, this form of forgetting remains particularly severe in MLLMs due to the complex interplay between modalities and the high dimensionality of the parameter space (Sung et al., 2023; Lin et al., 2023b; Luo et al., 2023; Zhai et al., 2024b; Shen et al., 2024; Jiang et al., 2024; Zhai et al., 2024a).

Existing approaches to mitigate catastrophic forgetting, such as experience replay (Riemer et al., 2018; Chaudhry et al., 2019) or regularization-based methods (Kirkpatrick et al., 2017), often incur high computational overhead or struggle to scale to the billions of parameters in MLLMs. Parameter-Efficient Fine-Tuning (PEFT) methods like LoRA (Hu et al., 2022) reduce the trainable parameter count but do not inherently prevent the erosion of pre-trained knowledge, as recent studies show they can still disrupt reliable upstream features (Biderman et al., 2024; Zhu et al., 2024). Consequently, a critical challenge arises: *How can we enable MLLMs to adapt effectively to downstream tasks while robustly preserving their pre-trained capabilities without prohibitive costs?*

Our work is primarily inspired by the principle of activation-aware parameter saliency, most notably demonstrated by Activation-aware Weight Quantization (AWQ) (Lin et al., 2024). AWQ established that a minority of "salient" weights—those associated with large activation magnitudes—carry the bulk of a model's functional information, and that protecting these weights is life-critical for maintaining performance under precision reduction. We extend this insight to the problem of catastrophic forgetting, hypothesizing that these salient parameters are similarly fundamental to the model's general capabilities. This perspective is further reinforced by recent findings in model merging, such as Nobari et al. (2025); Yao et al. (2025), demonstrating that activation-guided consensus can effectively resolve parameter space interference when merging specialized models fine-tuned on different tasks. By surgically identifying and freezing these high-saliency regions before downstream training, we ensure that the model's core upstream knowledge anchors remain intact, while the remaining "quiet" parameters are liberated to provide the necessary plasticity for specialization. Unlike previous weight-centric approaches that rely on static magnitude pruning (Han et al., 2015; Frankle & Carbin, 2019), our activation-based strategy captures

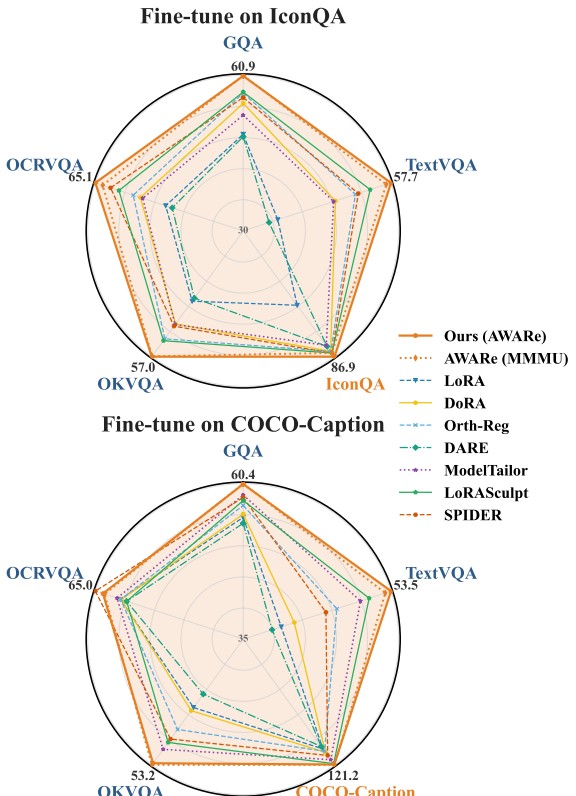

*Figure 2.* Performance comparison across multiple evaluation axes. The radar chart illustrates that AWARe achieves a superior balance between preserving upstream capabilities and adapting to downstream challenges compared to baseline regularization methods.

the dynamic, task-relevant importance of parameters.

Building on this motivation, we introduce **A**ctivation-**W**eighted **A**daptive **Re**taining (**AWARe**), a novel framework designed to surgically mitigate catastrophic forgetting. AWARe operates in two phases: *Knowledge Profiling* and *Constrained Fine-tuning*. First, in *Knowledge Profiling*, we utilize a small calibration set (e.g., 200 samples per upstream task) to compute activation-based saliency scores for network neurons. Notably, even in scenarios where specific upstream tasks are restricted, using a general-purpose dataset like MMMU (Yue et al., 2024) as a calibration source yields comparable efficacy. This profiling phase requires only a single forward pass over a few hundred samples, making its computational overhead negligible compared to full training. Second, during downstream training, we apply a mask freezing mechanism that locks these high-saliency parameters (Figure 1). This forces the optimization process to utilize only the less critical, plasticity-rich regions of the parameter space for the new task. Crucially, our method does not introduce any additional modules or architecture changes, maintaining the original inference efficiency.

We validate our approach through comprehensive experiments on standard visual question answering and captioning benchmarks. Our results demonstrate that AWARe achieves state-of-the-art trade-offs between stability and plasticity. Notably, by analyzing the activation landscape, we find that selectively freezing the top 30% of parameters within the self-attention mechanism—those identified as the most salient "knowledge anchors" for upstream tasks—is sufficient to achieve competitive downstream performance while virtually eliminating forgetting.

Our main contributions are summarized as follows:

❶ **AWARe Framework.** We introduce Activation-Weighted Adaptive REtaining (AWARe), a novel method that leverages activation patterns to dynamically identify and preserve critical upstream knowledge regions within MLLMs, achieving SOTA performance in balancing forgetting mitigation and downstream adaptation.

❷ **Efficiency and Simplicity.** Our approach requires no modification to the model architecture and operates by training only a subset of parameters (e.g., $\sim 17.5\%$; more detailed analysis is provided in Section B). It avoids the complexity of replay buffers or additional architectural components, making it highly efficient for large-scale deployment.

❸ **Comprehensive Validation.** We conduct a series of ablation studies to verify the effectiveness of activation-based saliency over weight-based metrics and explore the impact of retention ratios. Our empirical results confirm that AWARe robustly preserves generalization ability across diverse benchmarks.

## 2. Related Work

### 2.1. Multimodal LLM

Building upon the formidable reasoning capabilities of Large Language Models (LLMs) such as LLaMA (Touvron et al., 2023a;b) and Vicuna (Chiang et al., 2023), the research community has witnessed a paradigm shift towards Multimodal Large Language Models (MLLMs). These systems fundamentally aim to perceive and reason about visual signals as effectively as textual data. Pioneering frameworks like LLaVA (Liu et al., 2023) and MiniGPT-4 (Zhu et al., 2023) treat visual inputs as pseudo-tokens, projecting features extracted by strong visual encoders like CLIP (Radford et al., 2021) into the LLM's native embedding space. Diversity exists in the alignment mechanisms: while methodology like LLaVA adopts a concise Multi-Layer Perceptron (MLP) for projection, architectures such as BLIP-2 (Li et al., 2023) and InstructBLIP (Dai et al., 2023) introduce the Query Transformer (Q-Former) to compress visual representations. Further advancements have led to models with enhanced resolution and capability, including Qwen-VL series (Bai et al., 2023; Wang et al., 2024; Bai et al., 2025b;a), VILA (Lin et al., 2023a), and InternVL (Chen et al., 2024). Crucially, the paradigm of visual instruction tuning (Liu et al., 2023) empowers these models to generalize across unseen tasks by training on broad instruction-following data, although adapting them to specific vertical domains often necessitates further fine-tuning.

### 2.2. Catastrophic Forgetting in Multimodal LLM Fine-Tuning

Fine-tuning serves as a pivotal mechanism for adapting Multimodal Large Language Models (MLLMs) to downstream tasks. However, deep learning models often suffer from catastrophic forgetting (McCloskey & Cohen, 1989; McClelland et al., 1995), where previously learned knowledge is lost when acquiring new skills. Various continual learning algorithms have been proposed to address this issue, generally encompassing rehearsal-based, regularization-based (Kirkpatrick et al., 2017; Lopez-Paz & Ranzato, 2017), and architecture-based approaches (Houlsby et al., 2019; Lester et al., 2021). While established, many traditional methods rely on full-model fine-tuning or second-order statistics, making them computationally prohibitive for large-scale models.

In the era of large foundation models, the challenge shifts towards maintaining the model's strong generalization and "Open-World Stabilization" after downstream adaptation (Ni et al., 2023; Zhai et al., 2024b). Recent efforts have explored parameter-efficient solutions to mitigate forgetting. Reparameterization methods such as DoRA (Liu et al., 2024) and LoRA-based extensions like LoRAMoE (Dou et al., 2024) aim to enhance learning capacity while addressing generalization loss, whereas LoRASculpt (Liang et al., 2025) prunes redundant parameters via sparse updates. Regularization techniques have also been adapted; for instance, Orth-Reg (Hu et al., 2024) encourages fine-tuned features to remain orthogonal to pre-trained representations. CorDA (Yang et al., 2024) builds task-aware adapters through context-guided weight decomposition to preserve world knowledge. Selective tuning approaches, such as SPIDER (Huang et al., 2025) and Model Tailor (Zhu et al., 2024), leverage gradient or saliency analysis to construct sparse updates, aiming to protect critical pre-trained features. However, these criteria often rely on static information or gradient magnitudes, overlooking the dynamic nature of neuronal activations during forward propagation.

Recognizing these limitations, we propose Activation-Weighted Adaptive REtaining (AWARe). Unlike additive methods that introduce extra parameters or selective tuning that relies on static priors, AWARe utilizes task-induced activation distributions to measure parameter importance. By adaptively constraining high-activation neurons based on

*Figure 3.* Overview of the AWARe framework. (a) Phase 1 (Knowledge Profiling): We utilize activation distributions from upstream tasks to identify salient neurons. (b) Phase 2 (Constrained Fine-tuning): A mask freezing mechanism is applied to protect these salient regions, while allowing other parameters to adapt to downstream tasks.

upstream statistics, AWARe effectively balances the preservation of core capabilities with the flexibility needed for downstream task adaptation.

## 3. Methodology

### 3.1. Problem Formulation

A Multimodal Large Language Model (MLLM) $\mathcal{M}_\theta$ generally comprises three core components: a vision encoder $\mathcal{V}$ (e.g., CLIP (Radford et al., 2021) or SigLIP (Zhai et al., 2023)), a large language model $\mathcal{L}$ (e.g., LLaMA (Touvron et al., 2023b;a) or Vicuna (Chiang et al., 2023)), and a connector module $\rho$ that bridges the visual and textual modalities. The standard paradigm involves aligning the pre-trained vision encoder representations with the LLM's embedding space via the connector, enabling the model to process multimodal inputs effectively.

Consider such a VLM $\mathcal{M}_\theta$ (e.g., LLaVA (Li et al., 2024a)) parameterized by $\theta$. The model has been previously trained on a set of upstream tasks $\mathcal{T}_{up} = \{\text{OKVQA, OCRVQA, GQA, TextVQA}\}$. Our objective is to learn a single downstream task $\mathcal{T}_{down} \in \{\text{IconQA, COCO Caption}\}$ by updating $\theta$, while mitigating the *catastrophic forgetting* (McCloskey & Cohen, 1989; McClelland et al., 1995) of $\mathcal{T}_{up}$. As illustrated in Figure 3, we achieve this by partitioning the parameters into a fixed set $\theta_{fixed}$, which captures core upstream features, and a plastic set $\theta_{active}$ used for acquiring task-specific knowledge.

### 3.2. Activation-Based Saliency Estimation

To identify important parameters for retention, we utilize a calibration set $\mathcal{S}_{cal}$ composed of representative samples from each upstream task's train datasets ($\mathcal{T}_{up}$). Specifically, we randomly sample several instances from each up-

stream task. In scenarios where access to upstream data is restricted, we demonstrate that utilizing a comprehensive general benchmark, such as MMMU (Yue et al., 2024), as a proxy for $\mathcal{S}_{cal}$ is equally effective. We empirically demonstrate the robustness of model performance to the calibration set size in our ablation studies at Section 4.5. Crucially, the profiling incurs negligible overhead, requiring only a single forward pass on a small sample set. For a target linear layer with weight matrix $\mathbf{W} \in \mathbb{R}^{d_{out} \times d_{in}}$, we analyze its activation tensor $\mathbf{A} \in \mathbb{R}^{B \times L \times d_{out}}$, where $i \in \{1, \ldots, B\}$, $j \in \{1, \ldots, L\}$, and $k \in \{1, \ldots, d_{out}\}$ index the context length, batch size, and hidden dimension, respectively.

The saliency estimation follows a three-step aggregation and normalization process. First, we compute the $L_2$-norm of activations along the sequence length dimension for each sample $j$ and neuron $k$:

$$a'_{i,k} = \sqrt{\sum_{j=1}^{L} a^2_{i,j,k}} \tag{1}$$

Second, to ensure comparability across samples and prevent those with outlier-scale activations from biasing the estimations, we apply per-sample $L_2$-normalization across the hidden dimension:

$$a''_{i,k} = \frac{a'_{i,k}}{\|\mathbf{a}'_i\|_2} = \frac{a'_{i,k}}{\sqrt{\sum_{k'=1}^{d_{out}} {a'_{i,k'}}^2}} \tag{2}$$

where $\mathbf{a}'_i = [a'_{i,1}, \ldots, a'_{i,d_{out}}]^\top$ is the vector of sequence-aggregated activations for sample $i$. This normalization ensures that the saliency is determined by the *relative* importance of neurons within each context rather than absolute activation scale, which is crucial for identifying neurons that consistently capture structural features across diverse upstream tasks. This refinement distinguishes our approach from simple magnitude-based methods and explains the superior performance observed in Table 2. Finally, the saliency score for each neuron $k$ is obtained by averaging across the batch:

$$s_k = \frac{1}{B} \sum_{i=1}^{B} a''_{i,k} \tag{3}$$

The resulting saliency vector $\mathbf{s} \in \mathbb{R}^{d_{out}}$ reflects the relative importance of each output neuron in preserving upstream task capabilities.

### 3.3. Adaptive Parameter Retaining

Guided by the saliency scores, we introduce a retention ratio $\rho \in (0, 1)$ as a hyperparameter to control the fraction of parameters to be frozen. We identify the index set $\mathcal{I}$ containing the indices of the top $\rho \cdot d_{out}$ largest elements in $\mathbf{s}$ (e.g., top 30% or 10%). This selection is performed *independently per layer*, ensuring that parameter retention is calibrated to the specific feature distribution of each layer rather than applying a global threshold.

Since these salient neurons characterize the core upstream knowledge, we freeze their corresponding parameters in the weight matrix. Specifically, we construct a binary gradient mask $\mathbf{M}_{grad} \in \{0,1\}^{d_{out} \times d_{in}}$ such that:

$$(M_{grad})_{k,:} = \mathbf{0} \quad \text{if } k \in \mathcal{I}, \quad (M_{grad})_{k,:} = \mathbf{1} \quad \text{otherwise} \tag{4}$$

$$M_{ret} = \mathbf{1} - M_{grad} \tag{5}$$

where $\mathbf{0}$ and $\mathbf{1}$ are row vectors of size $d_{in}$. During downstream optimization, the parameters associated with salient regions are held constant via masked gradient updates:

$$\mathbf{W}^{(t+1)} = \mathbf{W}^{(t)} - \eta \cdot (\nabla_{\mathbf{W}} \mathcal{L}_{down} \odot \mathbf{M}_{grad}) \tag{6}$$

where $\eta$ is the learning rate and $\odot$ denotes the Hadamard product. In practice, the PyTorch implementation achieves this by utilizing a training-time wrapper, `AwareLinear`, which decomposes the linear layer into active and frozen components to efficiently manage gradient updates without altering the underlying operator's logic. This wrapper is removed post-training, leaving the original model configuration intact. (Implementation details are provided in Section C.) This row-wise freezing mechanism ensures that the most influential output features of the pre-trained LLaVA model remain intact while allowing less critical regions to adapt to new tasks. As shown in Figure 4, the distribution of these salient neurons varies across transformer layers and projection types.

### 3.4. Selection of Target Linear

We specifically apply AWARe to the linear projection layers within the self-attention mechanism (i.e., q_proj, k_proj, v_proj) and the linear layers within the multimodal projector (mm_projector). This selection is informed by recent research (Zhu et al., 2025b) which demonstrates that updating self-attention projections tends to cause significantly less catastrophic forgetting of pre-existing knowledge compared to the multilayer perceptron (MLP) blocks. By focusing our retention strategy on these linear, we balance the preservation of core cross-modal reasoning capabilities with the flexibility needed for downstream task adaptation. An ablation analysis is provided in Section 4.4. Consistent with this finding, all other components of the LLM, including MLP blocks and the output projection ($\mathbf{W}_o$), are kept entirely frozen during adaptation. This configuration not only maximizes parameter efficiency but also minimizes the risk of distorting the pre-trained feature space in regions less critical for downstream alignment. The complete procedure for AWARe is summarized in Algorithm 1.

### 3.5. Theoretical Insight

To provide a justification for AWARe, we analyze the representational shift from the perspective of *Feature Consistency*.

---

**Algorithm 1** AWARe

**input** Pre-trained model $\mathcal{M}_\theta$, Upstream tasks $\mathcal{T}_{up}$, Downstream task $\mathcal{T}_{down}$, Retention ratio $\rho$
**output** Fine-tuned model $\mathcal{M}_\theta$
 1: **Phase 1: Knowledge Profiling**
 2: Sample calibration set $\mathcal{S}_{cal} \sim \mathcal{T}_{up}$
 3: **for** each target layer $l$ in $\mathcal{M}_\theta$ **do**
 4:  Perform forward pass on $\mathcal{S}_{cal}$ to obtain activations $\mathbf{H}^{(l)}$
 5:  Calculate neuron saliency $\mathbf{s}^{(l)}$ using Equations (1) to (3)
 6:  $\mathcal{I}^{(l)} \leftarrow \text{TopIndices}(\mathbf{s}^{(l)}, \rho)$
 7:  Generate row-wise gradient mask $\mathbf{M}_{grad}^{(l)}$ via Equation (5)
 8: **end for**
 9: **Phase 2: Constrained Fine-tuning**
10: **while** not converged on $\mathcal{T}_{down}$ **do**
11:  $\mathcal{L} \leftarrow \text{ComputeLoss}(\mathcal{M}_\theta, \mathcal{T}_{down})$
12:  Update parameters: $\theta \leftarrow \theta - \eta \cdot (\nabla_\theta \mathcal{L} \odot \mathbf{M}_{grad})$
13: **end while**

---

While standard variance-based methods prioritize neurons with high absolute magnitudes, such an approach can be biased by outliers or scale differences across layers. Instead, AWARe's normalization step (Eq. 2) ensures we select neurons that are surprisingly salient relative to their context. We posit that "knowledge" is encoded in these consistently dominant relative directions.

### 3.6. Theoretical Insight

To justify AWARe, we analyze the representational shift through the lens of *Feature Consistency*. While standard magnitude-based pruning prioritizes neurons with high absolute activation, this can be biased by layer-specific scaling or outliers. Instead, AWARe seeks to preserve the "semantic skeleton" of the representation—the directions that consistently define the orientation of the embedding space.

**Lemma 3.1.** *(Directional Alignment Preservation) Let $\mathbf{y}_i \in \mathbb{R}^{d_{out}}$ be the output representation for sample $i$. To ensure the semantic orientation of the representation remains stable during adaptation, we aim to preserve neurons that contribute most to the direction of $\mathbf{y}_i$. We define the Directional Importance of neuron $k$ as its expected squared contribution to the unit direction vector: $s_k = \mathbb{E}[(\frac{y_{i,k}}{\|\mathbf{y}_i\|_2})^2]$.*

*Let $M_{ret} \in \{0,1\}^{d_{out}}$ be a binary retention mask where $m_k = 1$ if neuron $k$ is preserved (frozen). Maximizing the Preserved Directional Energy defined as:*

$$\mathcal{P}(M_{ret}) = \mathbb{E}\left[\frac{\|M_{ret} \odot \mathbf{y}_i\|_2^2}{\|\mathbf{y}_i\|_2^2}\right] \tag{7}$$

*is equivalent to selecting the top-$K$ neurons with the highest*

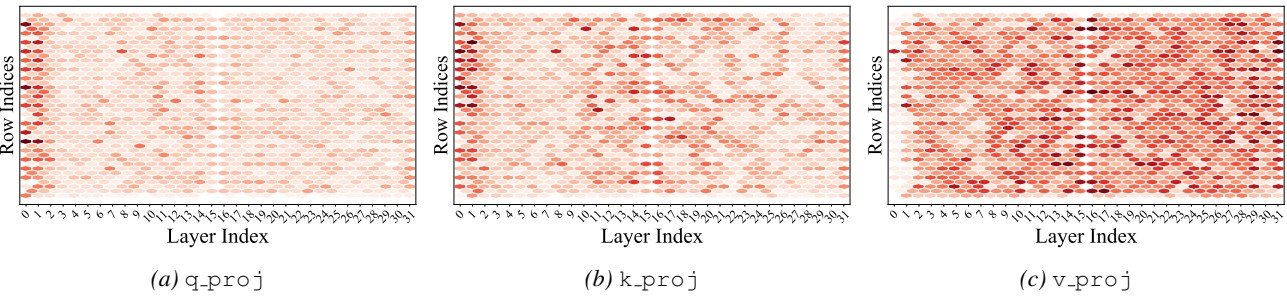

*(a)* `q_proj`       *(b)* `k_proj`       *(c)* `v_proj`

*Figure 4.* Layer-wise distribution of high-activation neurons across different projection types. The heatmaps visualize indices identified for freezing (top 30% global activation). Notably, `q_proj` and `k_proj` show high concentration in early layers, whereas `v_proj` exhibits a more uniform distribution, suggesting value information is distributed more evenly across the depth of the network.

*Directional Importance $s_k$.*

**Remark.** This aligns with "Hyperspherical Energy" learning (Liu et al., 2018), where angular relationships encode semantics. By anchoring these dominant relative directions, AWARe preserves the core alignment while channeling plasticity into less structurally critical dimensions.

## 4. Experiments

In this section, we evaluate the effectiveness of our proposed method by comparing it with several state-of-the-art baselines on multimodal downstream tasks. We also conduct extensive ablation studies to analyze the impact of key components and hyperparameters.

### 4.1. Experimental Setup

**Datasets and Tasks.** We utilize the LLaVA-v1.5-7b (Liu et al., 2023) model as our base model for training. This model has been pre-trained on several upstream tasks, including OKVQA (Marino et al., 2019), OCRVQA (Mishra et al., 2019), GQA (Hudson & Manning, 2019), and TextVQA (Singh et al., 2019). Our objective is to maximize performance on new downstream tasks while maintaining the model's capabilities on these upstream tasks to prevent catastrophic forgetting. For the downstream evaluation, we select two distinct tasks:

- **IconQA:** (Lu et al., 2021) A Visual Question Answering (VQA) task that requires abstract reasoning. We use accuracy (Acc) as the evaluation metric.
- **COCO-Caption:** (Lin et al., 2014) An image captioning task. We use the CIDEr score to measure the similarity between the generated captions and the ground truth.

For the upstream tasks, we report the accuracy to assess the extent of forgetting.

**Implementation Details.** During training, we freeze the vision tower to preserve the pre-trained visual representations. The learning rate is set to $1 \times 10^{-4}$ for the multimodal (MM)

projector and $1 \times 10^{-5}$ for the other trainable parameters. We train for 3 epochs using a cosine annealing learning rate scheduler with a warm-up ratio of 0.03. The total batch size is set to 16.

### 4.2. Baselines

We compare our method against several established parameter-efficient fine-tuning and continual learning baselines, including **Full-FT**, **LoRA** (Hu et al., 2022), **DoRA** (Liu et al., 2024), **DARE** (Yu et al., 2024), **Orth-Reg** (Hu et al., 2024), **Model Tailor** (Zhu et al., 2024), **LoRASculpt** (Liang et al., 2025), and **SPIDER** (Huang et al., 2025). Detailed descriptions and hyperparameter settings for all baselines are provided in Section E.

### 4.3. Main Results

We evaluate the performance of all methods by decomposing continual learning capabilities into two core components: **Knowledge Retention** ($\mathcal{R}$) and **Learning Efficiency** ($\mathcal{E}$). To provide a balanced assessment, we report their Harmonic Mean ($\mathcal{H}$):

$$\mathcal{H} = \frac{2 \cdot \mathcal{R} \cdot \mathcal{E}}{\mathcal{R} + \mathcal{E}} \qquad (8)$$

The components are defined as follows:

- **Knowledge Retention** ($\mathcal{R}$): A stability metric measuring the preservation of performance on upstream tasks (OKVQA, OCRVQA, GQA, TextVQA) after fine-tuning. It is calculated as the average ratio of the final accuracy to the base (zero-shot) accuracy:

$$\mathcal{R} = \frac{1}{|\mathcal{T}_{up}|} \sum_{i \in \mathcal{T}_{up}} \frac{Acc_{i,\text{final}}}{Acc_{i,\text{base}}} \qquad (9)$$

where $Acc_{i,\text{base}}$ is the model's initial performance before fine-tuning. A sharp drop in $\mathcal{R}$ indicates catastrophic forgetting.

- **Learning Efficiency** ($\mathcal{E}$): A plasticity metric measuring the model's adaptation to the new downstream task

Table 1. **Comparison with State-of-the-Art Fine-Tuning Solutions for Multimodal Large Language Models (MLLMs)** on visual question answering task IconQA and image captioning task COCO-Caption. The optimal and sub-optimal results are denoted by boldface and underlining. ↑ means improved accuracy compared with the sub-optimal results. †Specifically, the near-zero performance of Full-FT on upstream tasks reflects extreme catastrophic forgetting due to overfitting to the target task. More details about the metrics are provided in Section 4.3. Training setting are detailed in Section D.

| Methods | IconQA | | | | | | | | COCO-Caption | | | | | | | |
|---|---|---|---|---|---|---|---|---|---|---|---|---|---|---|---|---|
| | OKVQA | OCRVQA | GQA | TextVQA | *Target* | $\mathcal{R}$ | $\mathcal{E}$ | $\mathcal{H}$ | OKVQA | OCRVQA | GQA | TextVQA | *Target* | $\mathcal{R}$ | $\mathcal{E}$ | $\mathcal{H}$ |
| Zero-shot | 57.99 | 66.20 | 61.93 | 58.23 | 23.18 | 100.0 | 28.9 | 44.9 | 57.99 | 66.20 | 61.93 | 58.23 | 40.40 | 100.0 | 42.3 | 59.4 |
| Full-FT † | 0.04 | 0.00 | 0.00 | 0.46 | 80.15 | 0.2 | 100.0 | 0.4 | 0.00 | 0.00 | 0.00 | 0.04 | 95.59 | 0.0 | 100.0 | 0.0 |
| LoRA | 45.06 | 48.40 | 49.22 | 36.47 | 63.65 | 73.3 | 79.4 | 76.2 | 44.92 | 59.90 | 54.88 | 39.75 | 110.27 | 81.6 | 115.4 | 95.6 |
| DoRA | 49.94 | 54.40 | 55.39 | 47.26 | 84.48 | 84.7 | 105.4 | 93.9 | 45.38 | 59.75 | 55.40 | 41.38 | 112.60 | 82.6 | 117.8 | 97.1 |
| Orth-Reg | 53.12 | 56.10 | 57.43 | 51.00 | 84.52 | 89.1 | 105.5 | 96.6 | 48.11 | 59.85 | 56.83 | 46.69 | 111.87 | 86.5 | 117.0 | 99.5 |
| DARE | 44.39 | 46.85 | 48.75 | 34.84 | 82.28 | 71.5 | 102.7 | 84.3 | 42.98 | 58.65 | 53.92 | 38.61 | 108.57 | 79.5 | 113.6 | 93.5 |
| Model Tailor | 50.07 | 53.85 | 53.04 | 46.92 | 81.76 | 83.4 | 102.0 | 91.8 | 50.99 | 60.60 | 58.54 | 49.65 | 117.64 | **89.9** | 123.1 | 103.9 |
| LoRASculpt | 53.52 | 59.50 | 57.63 | 53.76 | 85.26 | 91.8 | 106.4 | 98.6 | 49.99 | 58.65 | 57.63 | 50.73 | 120.35 | 88.8 | 125.9 | 104.2 |
| SPIDER | 50.41 | 61.49 | 56.54 | 51.54 | 85.29 | 90.0 | 106.4 | 97.5 | 49.50 | 65.00 | 58.14 | 45.33 | 114.74 | 89.2 | 120.0 | 102.3 |
| AWARe | 56.95 | 65.10 | 60.83 | 57.66 | 86.92 | 98.4 | 108.4 | 103.2↑4.6 | 53.01 | 63.45 | 60.32 | 53.46 | 120.94 | 94.2 | 126.5 | 108.0↑3.8 |
| *AWARe (MMMU)* | 56.86 | 63.35 | 60.93 | 56.77 | 85.23 | 97.4 | 106.3 | 101.7↑3.1 | 53.25 | 63.12 | 60.38 | 52.74 | 121.22 | 93.9 | 126.8 | 107.9↑3.7 |

(IconQA or COCO-Caption). To normalize against model capacity, it is defined relative to the Full Fine-Tuning (Full-FT) baseline:

$$\mathcal{E} = \frac{Acc_{down,\text{Method}}}{Acc_{down,\text{Full-FT}}} \quad (10)$$

where $Acc_{down,\text{Full-FT}}$ represents the performance upper bound achievable by fully updating all parameters.

As shown in Table 1, our method achieves a superior harmonic balance $\mathcal{H}$ between plasticity and stability. Compared to other methods, our approach maintains high retention rates $\mathcal{R}$ while achieving efficiency $\mathcal{E}$. Notably, when upstream task data is restricted, we used the MMMU general dataset as a calibration dataset and still achieved relatively good results, outperforming both LoRASculpt and SPIDER.

### 4.4. Ablation Studies

We conduct ablation studies to investigate the contribution of different components and hyperparameter choices in our framework.

**Effectiveness of Activation-based Selection.** We compare our activation-based parameter selection strategy against weight norm-based selection, a hybrid strategy ($0.3 \times$ Weight $+ 0.7 \times$ Activation), and a baseline using random parameter selection at the same ratio (30%). As shown in Table 2, random selection significantly fails to protect the upstream knowledge, with the average performance on upstream tasks (*Source Avg*) dropping to 49.94 on the IconQA task. In contrast, our activation-based selection consistently achieves the best balance, outperforming both random and weight-based selection in both *Source Avg* and *Target* scores. Specifically, the activation-guided approach preserves the highest upstream performance while achieving superior adaptation on downstream tasks, validating that high-activation neurons are indeed more critical for knowledge retention.

Table 2. **Ablation Study on Parameter Selection Strategy.** We compare different methods for selecting parameters to retain: random selection (Random Selection), weight magnitude (Weight Norm), activation magnitude (Activation), and a hybrid approach ($0.3W + 0.7A$).

| Strategy | IconQA | | COCO-Caption | |
|---|---|---|---|---|
| | *Source Avg* | *Target* | *Source Avg* | *Target* |
| Random Selection | 49.94 | 86.32 | 49.46 | 116.31 |
| Weight Norm | 58.51 | 85.51 | 53.52 | 115.19 |
| $0.3W + 0.7A$ | 58.69 | 84.66 | 53.90 | 115.24 |
| **Activation** | **60.13** | **86.92** | **57.56** | **120.94** |

Table 3. **Ablation Study on Selection Ratio.** We evaluate the performance of our method with different ratios of parameters selected for retention on the IconQA task.

| Ratio | OKVQA | OCRVQA | GQA | TextVQA | *Target* | $\mathcal{R}$ | $\mathcal{E}$ | $\mathcal{H}$ |
|---|---|---|---|---|---|---|---|---|
| *Layer balanced selection* | | | | | | | | |
| 1% | 55.33 | 64.00 | 60.17 | 54.39 | 82.44 | 95.7 | 102.9 | 99.2 |
| 10% | 55.33 | 65.20 | 59.64 | 55.16 | 86.84 | 96.3 | 108.3 | 102.0 |
| 30% | 56.28 | 62.75 | 60.60 | 56.13 | 85.29 | 96.5 | 106.4 | 101.2 |
| 40% | 56.47 | 63.85 | 60.46 | 56.94 | 86.13 | 97.3 | 107.5 | 102.1 |
| 50% | 57.33 | 63.95 | 60.80 | 57.19 | 86.21 | 97.9 | 107.6 | 102.5 |
| 70% | 57.69 | 64.20 | 61.77 | 57.54 | 85.67 | 98.7 | 106.9 | 102.6 |
| 90% | 57.65 | 63.70 | 62.01 | 57.92 | 84.07 | 98.7 | 104.9 | 101.7 |
| *Global highest selection* | | | | | | | | |
| 1% | 55.26 | 63.40 | 59.89 | 54.96 | 82.65 | 95.6 | 103.1 | 99.2 |
| 10% | 55.88 | 63.25 | 59.59 | 55.39 | 86.13 | 95.8 | 107.5 | 101.3 |
| **30%** | 56.95 | 65.10 | 60.83 | 57.66 | **86.92** | 98.4 | 108.4 | **103.2** |
| 40% | 57.66 | 64.35 | 61.47 | 57.77 | 86.64 | 98.7 | 108.1 | **103.2** |
| 50% | 57.71 | 64.75 | 61.87 | 57.86 | 85.97 | 99.1 | 107.3 | 103.0 |
| 70% | 57.78 | 64.35 | 62.03 | 57.84 | 80.64 | 99.0 | 100.6 | 99.8 |
| 90% | **58.13** | **65.55** | 61.89 | **58.33** | 54.62 | **99.8** | 68.1 | 81.0 |

**Impact of Selection Ratio and Strategy.** We explore the effect of varying the selection ratio (from 1% to 90%) and comparing Layer-Balanced versus Global-Highest selection strategies. As shown in Table 3, the Global-Highest selection strategy at a 30% ratio achieves the optimal balance between stability and plasticity, yielding the highest harmonic mean ($\mathcal{H} = 103.2$) and superior knowledge retention ($\mathcal{R} = 98.4$).

**Calibration Dataset Composition.** We examine how the composition of the calibration dataset influences the model's

385
386
387
388
389
390
391
392
393
394
395
396
397
398
399
400
401
402
403
404
405
406
407
408
409
410
411
412
413
414
415
416
417
418
419
420
421
422
423
424
425
426
427
428
429
430
431
432
433
434
435
436
437
438
439

*Table 4.* **Ablation Study on Calibration Dataset Composition.** We investigate how using different subsets of upstream data for activation statistics calculation affects performance on the IconQA task.

| Calibration Task | OKVQA | OCRVQA | GQA | TextVQA | *Target* | $\mathcal{R}$ | $\mathcal{E}$ | $\mathcal{H}$ |
|---|---|---|---|---|---|---|---|---|
| OKVQA | 55.17 | 63.96 | 58.56 | 54.72 | **86.97** | 95.1 | **108.5** | 101.4 |
| OCRVQA | 54.76 | **65.16** | 58.28 | 54.45 | 85.85 | 95.2 | 107.1 | 100.8 |
| GQA | 55.92 | 64.22 | 60.12 | 54.97 | 86.21 | 96.3 | 107.6 | 101.6 |
| TextVQA | 55.56 | 64.40 | 59.78 | 56.11 | 85.78 | 96.5 | 107.0 | 101.5 |
| ALL | **56.95** | 65.10 | **60.83** | **57.66** | 86.92 | **98.4** | 108.4 | **103.2** |

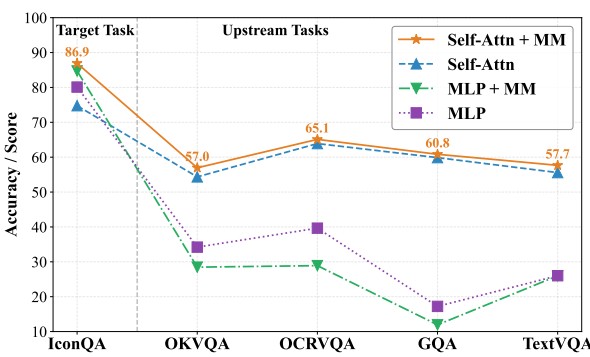

*Figure 5.* **Ablation Study on Training Target.** Performance comparison across different architecture components on the IconQA and upstream tasks. Applying AWARe to both `self-attn` and `mm_projector` achieves the best stable-plastic trade-off.

resistance to catastrophic forgetting on the IconQA task. As shown in Table 4, using any individual upstream task (e.g., OKVQA, OCRVQA, GQA, or TextVQA) as the calibration source consistently preserves knowledge, with relatively small performance variations across different configurations. However, our comprehensive calibration approach (ALL), which integrates samples from all upstream sources, yields the most balanced and superior overall performance ($\mathcal{H} = 103.2$). This demonstrates that while AWARe is robust to the specific choice of calibration task, a diverse mixture of upstream data ensures the most effective protection across the model's entire functional landscape.

**Training Targets.** We investigate the impact of applying our method to different components within the model architecture. As illustrated in Figure 5, targeting both the self-attention projections and the multimodal (`mm_projector`) yields the most favorable balance between plasticity and stability. Specifically, the `mm_projector` is essential for learning new downstream concepts; Locking it limits adaptation and causes a roughly 10-point decline in target performance. Furthermore, we observe that fine-tuning `MLP` blocks results in a severe performance collapse on upstream tasks (dropping below 30-point), even with our adaptive protection. This reinforces the hypothesis that MLP weights house the bulk of the model's fundamental general knowledge, whereas self-attention projections provide a more flexible landscape for specialized task adaptation.

*Table 5.* **Sensitivity Analysis on Calibration Datasets.** We evaluate the performance of AWARe on IconQA task across different calibration set sizes (200 and 400) and random sample 3 times.

| Calibration Size | OKVQA | OCRVQA | GQA | TextVQA | *Target* | $\mathcal{R}$ | $\mathcal{E}$ | $\mathcal{H}$ |
|---|---|---|---|---|---|---|---|---|
| 200 | 56.95 | 65.10 | 60.83 | 57.66 | 86.92 | 98.4 | 108.4 | 103.2 |
| 200 | 56.88 | 64.45 | 61.20 | 58.04 | 86.57 | 98.5 | 108.0 | 103.0 |
| 200 | 56.49 | 64.50 | 60.49 | 58.15 | 86.58 | 98.1 | 108.0 | 102.8 |
| 400 | 56.37 | 64.38 | 60.58 | 57.77 | 86.20 | 97.9 | 107.5 | 102.5 |
| 400 | 56.74 | 64.40 | 60.38 | 57.17 | 86.69 | 97.7 | 108.2 | 102.7 |
| 400 | 57.04 | 64.62 | 60.47 | 56.77 | 86.35 | 97.8 | 107.7 | 102.5 |
| *Std. Dev.* | *0.24* | *0.25* | *0.28* | *0.48* | *0.23* | *0.30* | *0.30* | *0.30* |

### 4.5. Sensitivity to Calibration Datasets

To assess the statistical significance and robustness of our results, we evaluate the sensitivity of AWARe to the randomness of the calibration set. We repeat the Knowledge Profiling phase over three independent runs with different random seeds for sample selection. As shown in Table 5, the performance metrics exhibit minimal standard deviation across runs, demonstrating that the identified salient neurons are consistent and statistically stable artifacts of the model structure, rather than noise from specific data samples. Moreover, increasing the calibration size from 200 to 400 samples yields negligible performance differences, confirming that 200 samples are statistically sufficient to robustly estimate parameter importance.

### 5. Conclusion

In this paper, we introduce AWARe, an activation-weighted parameter-efficient framework designed to address catastrophic forgetting in Multimodal Large Language Models (MLLMs). Our method consists of two primary phases: *Knowledge Profiling*, which identifies salient neurons based on task-induced activation distributions from upstream tasks, and *Constrained Fine-tuning*, which selectively freezes these critical regions to protect pre-trained capabilities. AWARe offers three distinct advantages: First, **Architecture Agnosticism**, as it operates without altering the model structure or requiring additional parameters like adapters. Second, **High Parameter-Efficiency**, effectively adapting models by updating only a small subset (e.g., 17.5%) of the parameters. Third, **Simplicity and Effectiveness**, providing a simple yet powerful solution that achieves exceptional performance within upstream and downstream tasks. Notably, we demonstrate that AWARe remains highly effective even when using a general-purpose dataset like MMMU for profiling, confirming its robustness in data-scarce or task-agnostic scenarios. Through rigorous theoretical analysis and extensive experiments, we demonstrate that AWARe provides a superior balance between stability and plasticity, offering a scalable solution for the sustainable deployment of MLLMs.

## 6. Impact Statement

This paper presents work whose goal is to advance the field of Machine Learning. There are many potential societal consequences of our work, none which we feel must be specifically highlighted here.

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

## A. Proofs

### A.1. Proof of Lemma 3.1

*Proof.* Let $\hat{\mathbf{y}}_i = \frac{\mathbf{y}_i}{\|\mathbf{y}_i\|_2}$ denote the unit direction vector of the output representation for sample $i$. The Preserved Directional Energy measures how much of the unit vector's magnitude is retained after masking. Using the definition of the $\ell_2$ norm, we expand the objective function:

$$\mathcal{P}(M_{ret}) = \mathbb{E}\left[\|M_{ret} \odot \hat{\mathbf{y}}_i\|_2^2\right] \tag{11}$$

$$= \mathbb{E}\left[\sum_{k=1}^{d_{out}} ((M_{ret})_k \hat{y}_{i,k})^2\right] \tag{12}$$

Since $M_{ret}$ is a binary vector, $(M_{ret})_k^2 = (M_{ret})_k$. By linearity of expectation, we can swap the summation and expectation:

$$\mathcal{P}(M_{ret}) = \sum_{k=1}^{d_{out}} (M_{ret})_k \cdot \mathbb{E}\left[\hat{y}_{i,k}^2\right] \tag{13}$$

Substituting the definition of Directional Importance $s_k = \mathbb{E}[\hat{y}_{i,k}^2]$, the optimization problem for a fixed budget $K$ becomes:

$$\max_{M_{ret}} \sum_{k=1}^{d_{out}} (M_{ret})_k s_k \quad \text{s.t.} \quad \|M_{ret}\|_0 = K, \quad (M_{ret})_k \in \{0,1\} \tag{14}$$

This is a linear knapsack problem where weights are equal. The global maximum is obtained by a greedy selection strategy: setting $(M_{ret})_k = 1$ for the indices $k$ corresponding to the $K$ largest values of $s_k$. Thus, preserving neurons with the highest expected normalized squared activation strictly maximizes the preserved directional energy of the representation. □

## B. Parameter Efficiency Analysis

In this section, we provide a detailed calculation of the trainable parameter ratio in our optimal setting. Consider a standard Transformer block in LLaMA architecture with hidden dimension $h$. The parameters can be categorized into Self-Attention and Feed-Forward Network (FFN) modules:

- **Self-Attention:** Consists of four projection matrices: $\mathbf{W}_q, \mathbf{W}_k, \mathbf{W}_v, \mathbf{W}_o \in \mathbb{R}^{h \times h}$. The total parameter count is $4h^2$.

- **FFN (SwiGLU):** Comprises three matrices: $\mathbf{W}_{up}, \mathbf{W}_{gate} \in \mathbb{R}^{h_{inter} \times h}$ and $\mathbf{W}_{down} \in \mathbb{R}^{h \times h_{inter}}$. In LLaMA, the intermediate dimension is typically set to $h_{inter} \approx \frac{8}{3}h$. The total parameter count is approximately $3 \times \frac{8}{3}h \times h = 8h^2$.

Thus, the total number of parameters per block is approximately $12h^2$.

In our best experimental setting, we apply the AWARe strategy specifically to the query, key, and value projection layers ($\mathbf{W}_q, \mathbf{W}_k, \mathbf{W}_v$) with a retention ratio of $\rho = 30\%$, while freezing the remaining components. The number of trainable parameters is calculated as:

$$N_{train} = 3 \times (1 - \rho) \times h^2 = 3 \times 0.7 \times h^2 = 2.1h^2 \tag{15}$$

Consequently, the ratio of trainable parameters to the total parameter count is:

$$\text{Ratio} = \frac{2.1h^2}{12h^2} \approx 17.5\% \tag{16}$$

This analysis demonstrates that our method effectively updates only a quarter of the model parameters, significantly reducing computational overhead while preserving generalization capabilities.

## C. Implementation Details of AWARe Linear Layer

In order to efficiently implement the row-wise freezing mechanism described in the methodology, we define a custom `AwareLinear` module. This module splits the original linear layer into two separate linear layers: one for the "active"

(trainable) neurons and one for the "frozen" (fixed) neurons. This separation avoids the computational overhead of applying large binary masks during forward passes and simplifies gradient management. The output is then reconstructed by gathering the partial outputs back into their original indices.

The PyTorch implementation is provided below:

```python
import torch
import torch.nn as nn

class AwareLinear(nn.Module):
    def __init__(self, original_linear: nn.Linear, freeze_pos):
        super().__init__()
        self.out_features = original_linear.out_features
        self.in_features = original_linear.in_features

        # Sort indices to maintain deterministic interactions
        self.frozen_pos = sorted(freeze_pos)
        self.active_pos = [
            i for i in range(self.out_features) if i not in self.frozen_pos
        ]

        # Validation checks
        if not all(0 <= idx < self.out_features for idx in self.active_pos):
            raise ValueError(
                f"Target neuron indices must be within [0, {self.out_features - 1}]"
            )
        if len(self.frozen_pos) != len(set(self.frozen_pos)):
            raise ValueError("Frozen neuron indices contain duplicate values")

        # Initialize sub-layers
        # 'active' handles trainable parameters
        # 'frozen' handles fixed parameters
        if original_linear.bias is not None:
            self.active = nn.Linear(self.in_features, len(self.active_pos), bias=True)
            self.frozen = nn.Linear(self.in_features, len(self.frozen_pos), bias=True)
        else:
            self.active = nn.Linear(self.in_features, len(self.active_pos), bias=False)
            self.frozen = nn.Linear(self.in_features, len(self.frozen_pos), bias=False)

        # Copy weights from the original model
        W = original_linear.weight.data
        self.active.weight = nn.Parameter(W[self.active_pos].clone(), requires_grad=True)
        self.frozen.weight = nn.Parameter(W[self.frozen_pos].clone(), requires_grad=False)

        if original_linear.bias is not None:
            B = original_linear.bias.data
            self.active.bias = nn.Parameter(B[self.active_pos].clone(), requires_grad=True
                ↪ )
            self.frozen.bias = nn.Parameter(B[self.frozen_pos].clone(), requires_grad=
                ↪ False)

        # Pre-compute index mapping for scatter/gather operations
        # This maps the concatenated output [active, frozen] back to original indices
        index_map = torch.empty(self.out_features, dtype=torch.long)
        index_map[self.active_pos] = torch.arange(len(self.active_pos))
        index_map[self.frozen_pos] = (
            torch.arange(len(self.frozen_pos)) + len(self.active_pos)
        )
        self.register_buffer("index_map", index_map)

    def forward(self, x: torch.Tensor) -> torch.Tensor:
        # Compute outputs separately
        active_out = self.active(x)
        frozen_out = self.frozen(x)
```

```
715    # Concatenate results
716    output = torch.cat([active_out, frozen_out], dim=-1)
717
718    # Reorder to match original output dimensions
719    return output.gather(
720        dim=-1,
721        index=self.index_map.to(x.device).expand_as(output),
722    )
```

*Listing 1.* PyTorch Implementation of AWARe Linear Layer

## D. Best Training Settings

We summarize the hyperparameter configuration that yields the optimal performance reported in our main results, specifically the **Global-Highest 30%** setting.

| Hyperparameter | Value |
|---|---|
| *Knowledge Profiling Phase* | |
| Calibration Dataset Composition | OKVQA, OCRVQA, GQA, TextVQA (200 samples each) |
| *When Upstream Dataset is restricted* | MMMU (20 samples each category 600 samples in total) |
| Selection Strategy | Global Highest Activation |
| Training Target Components | q_proj, k_proj, v_proj, mm_projector |
| Retention Quota ($\rho$) | 30% (Top 30% salient neurons frozen) |
| *Downstream Fine-tuning Phase* | |
| Epochs | 3 |
| Global Batch Size | 16 |
| Learning Rate (LLM Components) | $2 \times 10^{-5}$ |
| Learning Rate (MM Projector) | $2 \times 10^{-4}$ |
| LR Scheduler | Cosine Annealing (Warmup ratio: 0.03) |
| Optimizer | AdamW |

*Table 6.* Detailed Hyperparameter Settings for the Optimal AWARe Configuration.

## E. Baseline Details

We calculate baselines using the following configurations:

- **Full-FT**: Full fine-tuning of the model parameters.
- **LoRA**[ICLR'22](Hu et al., 2022): Low-Rank Adaptation, which injects trainable rank decomposition matrices.
- **DoRA**[ICML'24](Liu et al., 2024): Enhances LoRA's learning capacity and training stability by decomposing weights into magnitude and direction components.
- **DARE**[ICML'24](Yu et al., 2024): Parameters from the fine-tuned model are randomly selected and rescaled to preserve both generalization and specialization capabilities.
- **Orth-Reg**[ECCV'24](Hu et al., 2024): Encourages fine-tuned features to remain orthogonal to pretrained features, thereby preserving model generalization.
- **Model Tailor**[ICML'24](Zhu et al., 2024): Pre-trained parameters are preserved while a small fraction (e.g., 10%) of fine-tuned parameters is replaced based on salience and sensitivity analysis.
- **LoRASculpt**[CVPR'25](Liang et al., 2025): The method prunes redundant LoRA parameters via sparse updates guided by weight importance and conflict-aware regularization.
- **SPIDER**[ICML'25](Huang et al., 2025): Updates parameters only where specialization knowledge is more critical than generalization knowledge.

For all LoRA-based methods including **LoRA**, **DoRA**, and **LoRASculpt**, we set the adapter rank to $r = 32$. For methods involving parameter selection or masking, such as **DARE** and **Model Tailor**, we use the optimal selection ratio ($\rho$) to ensure

a fair comparison with our framework. Notably, baseline results for certain benchmarks are directly adopted from the original evaluations reported in LoRASculpt (Liang et al., 2025) and SPIDER (Huang et al., 2025) to ensure consistency and accuracy in our comparative analysis.

