# OpenReview forum: "AWARe: Mitigating Catastrophic Forgetting via Activation-Weighted Adaptive REtaining"
_ICML.cc/2026/Conference — Submitted to ICML 2026_

### Official Review · Reviewer_GKwX · 2026-02-26

**Soundness:** 2
**Presentation:** 3
**Significance:** 2
**Originality:** 2
**Overall Recommendation:** 3
**Confidence:** 4

**Summary:**

This paper proposes a continual learning method for MLLMs, named Activation-Weighted Adaptive REtaining (AWARe), a fine-tuning method that mitigates catastrophic forgetting by dynamically controlling parameter updates based on activation patterns. AWARe assigns activation-based importance scores to parameters, selectively freezing those essential for preserving prior capabilities while allowing less important parameters to adapt to new tasks. Extensive experiments demonstrate that AWARe effectively preserves upstream capabilities while achieving superior downstream performance compared to existing methods. The experiments use LlaVA-v1.5-7b as backbone, pretrained on several upsteam tasks, and test on two downstreams tasks IconVA and COCO-Caption.

**Compliance With Llm Reviewing Policy:**

Affirmed.

**Final Justification:**

The rebuttal solved my questions, but it remains that the novelty is not strong. So, I keep my recommendation of weak reject.

**Key Questions For Authors:**

Theretical insight (two subsections 3.5/3.6 with the same title):
Though Eq (7) shows that selecting the top parameters of directional importance maximizes the preserve energy, tthe relation of this energy with the generalization performance of model is not clear.

**Limitations:**

Not discussed in the paper.

**Strengths And Weaknesses:**

Strnegths:
1. A method carefully designed to improve the continual learning performance of MLLMs.
2. Well-organized and wrtitten paper.
Weaknesses:
1. Limited novelty of the method, which is amongh a set of similar methods based on selective parameter updating, but proposes a different parameter saliency estimation technique.
2. Insufficient experiments: only evaluated with one backbone MLLM, on only two downstream tasks. Unlike many other continual learning methods that usually evaluate on a series of downstream tasks.
3. Insufficient comparison with related methods. Here are two related works addressing the same MLLM continual learning:
H. Guo, et al.,  HiDe-LLaVA: Hierarchical Decoupling for Continual Instruction Tuning of Multimodal Large Language Model, ACL 2025.
W. Liu, et al., LLaVA-c: Continual Improved Visual Instruction Tuning, arXiv2506.08666, 13 Jun 2025.

---

> ### Author Rebuttal · Authors · 2026-03-31
>
> Thank you for acknowledging to our method. We provide explanations to your questions point-by-point in the following.
>
> ### ${\color{#f26921}\text{Weakness-1:}}$
> **Response:** We agree AWARe is not intended to define a new fine-tuning family. Our novelty lies in how to identify and preserve knowledge-critical parameters during adaptation. AWARe is motivated by the hypothesis that pretrained knowledge concentrates in neurons consistently dominant under upstream-task activations. It uses normalized task-induced forward activations not weight magnitude or gradients to locate these anchors.To the best of our knowledge, AWARe is the first to transfer AWQ-style activation-aware saliency to catastrophic-forgetting-aware fine-tuning in MLLMs. Just as AWQ protects quantization-sensitive weights, AWARe repurposes activation statistics to preserve capabilities during downstream adaptation.
>
> ### ${\color{#f26921}\text{Weakness-2:}}$
> **Response:** Our experiments follow **LoRASculpt** and **SPIDER**, focusing on **downstream fine-tuning with preservation of pretrained capabilities**.
>
> Nevertheless, appreciate the reviewerwer’s suggestion, we evaluated AWARe on a sequence of downstream tasks. The newly added continual-learning tests during rebuttal show strong performance. Importantly, unlike many continual-learning methods, AWARe requires **no architectural changes** and **no additional trainable parameters**.
>
> For **Qwen2.5-VL-7B**, see our response to **Reviewer QhMC (Weakness 1)**. For continual learning, we evaluated **LLaVA-v1.5** on **MLLM-DCL**; see **Reviewer NcH2 (Question 2)**.
>
> ### ${\color{#f26921}\text{Weakness-3:}}$
> **Response:** We agree continual-learning baselines are important. We evaluated AWARe on the **MLLM-DCL** benchmark and added **HiDe-LLaVA** as our primary continual-learning baseline due to its strong relevance. See **Reviewer NcH2 (Question 2)** for details.
>
> For **LLaVA-c**, although the paper states the code will be released, an official open-source implementation was inaccessible during the rebuttal, so we could not include it as a controlled baseline in the same manner.
>
> ### ${\color{#f26921}\text{Question-1:}}$
>
> > Theretical insight (two subsections 3.5/3.6 with the same title): Though Eq (7) shows that selecting the top parameters of directional importance maximizes the preserve energy, tthe relation of this energy with the generalization performance of model is not clear.
>
> **Response:** Thank you for point out, we will fix the duplicated subsection titles.
>
> We clarify that Eq. (7) is **not** intended as a full theorem for end-task generalization. Instead, it offers a **representation-stability justification** for our rule, corresponding to the retention metric $\mathcal{R}$ in Eq. (9) (preservation of upstream capabilities).
>
> Appendix A shows that for a fixed retention budget, selecting the top-K neurons by directional importance maximizes the **Preserved Directional Energy** in Eq. (7). For sample i, let:
>
> $$
> P_i(M_{\mathrm{ret}})=\frac{\left\\|M_{\mathrm{ret}}\odot y_i\right\\|_2^2}{\left\\|y_i\right\\|_2^2}
> $$
>
> $$
> \cos\big(y_i, M_{\text{ret}}\odot y_i\big)
> = \frac{y_i^\top (M_{\text{ret}}\odot y_i)}{\lVert y_i\rVert_2\lVert M_{\text{ret}}\odot y_i\rVert_2}
> = \frac{\lVert M_{\text{ret}}\odot y_i\rVert_2}{\lVert y_i\rVert_2}
> = \sqrt{P_i(M_{\text{ret}})}
> $$
>
> This quantity directly determines the angular distortion of the normalized representation:
>
> $$
> \left\\|\hat y_i - \widehat{M_{\mathrm{ret}}\odot y_i} \right\\|^2
> = 2\bigl(1-\sqrt{P_i(M_{\mathrm{ret}})}\bigr)
> $$
>
> Thus, maximizing preserved directional energy is equivalent to minimizing directional drift after masking, explaining why our rule best preserves the geometry of the pre-trained feature space under a fixed masking budget.
>
> The link to model performance is therefore through **prediction stability**, rather than a full statistical generalization guarantee. If the downstream readout is locally Lipschitz, smaller feature drift leads to smaller logit drift.
>
> $$
> \\|\hat y_i-\widehat{M_{\text{ret}}\odot y_i}\\|_2 < \gamma_i/L.
> $$
>
> When this perturbation remains below the prediction margin, the prediction is preserved. In this sense, Eq. (7) should be viewed as a principled **proxy for upstream retention**, not a direct guarantee of downstream generalization.
>
> That said, we agree that this theory is only **supporting evidence**, not a complete explanation of downstream generalization or the full stability–plasticity trade-off. Our main justification is empirical: Table 2 shows that activation-based selection outperforms weight-based and random alternatives; Table 3 shows the expected retention/adaptation trade-off as the retention ratio varies; and Tables 4 and 5 further show robustness across calibration sources and calibration set sizes. We will revise the paper to clarify this scope.
>
> **Thank you for your thoughtful comments. We hope our response addresses your concerns. Please let us know if you have any further questions.**

---

> > ### Author Rebuttal · Reviewer_GKwX · 2026-04-03
> >
> > The rebuttal solved my questions, but it remains that the novelty is not strong. So, I keep my recommendation of weak reject.

---

### Official Review · Reviewer_qF8R · 2026-03-12

**Soundness:** 3
**Presentation:** 3
**Significance:** 3
**Originality:** 2
**Overall Recommendation:** 4
**Confidence:** 4

**Summary:**

In this paper, the authors try to tackle catastrophic forgetting in multimodal LLM fine-tuning. The main framework, AWARe, computes activation-based saliency scores on a small calibration set from upstream tasks. Then it freezes the most salient rows of selected linear layers during downstream fine-tuning. The method aims to preserve upstream capability of an LLM while still allowing adaptation to a downstream task, without adding adapters or changing inference-time architecture. The experimental setup includes LLaVA-7B, (v1.5), with upstream evaluation on OKVQA, OCRVQA, GQA, and TextVQA, while the downstream datasets include IconQA, and COCO-Caption. The reported results on this setup indicates a better retention/adaptation trade-off than LoRA (and its variants), Orth-Reg, DARE, Model Tailor, and SPIDER.

**Compliance With Llm Reviewing Policy:**

Affirmed.

**Final Justification:**

The rebuttal took care of most of my questions/weaknesses.

**Key Questions For Authors:**

1. Can you evaluate AWARe on more base model families or at least one stronger alternative MLLM architecture beyond LLaVA-v1.5-7B?
2. Can you report results on additional downstream tasks, especially tasks that are less similar to the current setup, to test whether the method generalizes beyond IconQA and COCO-Caption?
3. How sensitive are the results to baseline hyperparameter tuning, especially for strong competitors like SPIDER, LoRASculpt, Orth-Reg, and Model Tailor?
4. Have you tested whether AWARe still works well in a more sequential setting with multiple downstream fine-tuning stages, rather than a single downstream adaptation step?

**Limitations:**

yes

**Strengths And Weaknesses:**

**Strengths**
- Catastrophic forgetting in MLLM fine-tuning is a timely topic of discussion.
- The introduced method (AWARe) is quite simple and hence easy to understand/implement. Simply using activation statistics from a small calibration set to identify "important" neurons, then masking their updates during fine-tuning. Does not require adding modules or changing anything in inference.
- The empirical results are reasonably strong on the paper's chosen setup as mentioned in the summary.
- I personally think that the ablation portion of the paper is the strongest part. The authors test activation-based selection against weight-based, random, and hybrid alternatives, vary retention ratio, vary calibration data composition, and finally study the sensitivity to calibration set size. This makes the paper more credible.
- It is essentially a parameter-efficient method at the end of the day.

**Weaknesses**
- Overall the scope of the evaluation is quite narrow. The entire study appears to use a single base model family, two downstream tasks, and a fairly specific upstream/downstream setup.
- The paper frames the method as broadly mitigating catastrophic forgetting, but the actual setting is much closer to one-step specialization while preserving pre-trained capabilities, not continual learning in a stronger sense.
- The main metric design is somewhat debatable. I would have liked more emphasis on raw downstream performance and forgetting deltas alongside the composite metric.
- The theoretical justification is fairly weak relative to how confidently it is presented. The “Directional Alignment Preservation” lemma mainly formalizes that if one wants to preserve directional energy, selecting the top contributing neurons is optimal under that objective. Intuitively that is fine. But it is not a deep justification for why this exact saliency measure should best preserve upstream multimodal knowledge in practice.
- The results are promising, but I do not think two downstream tasks on one model family are enough to support a broad SOTA claim for MLLM forgetting mitigation.

---

> ### Author Rebuttal · Authors · 2026-03-31
>
> Thank you for acknowledging to our method. We provide explanations to your questions point-by-point in the following.
>
> ### ${\color{#f26921}\text{Weakness-1:}}$
>
> **Response**:
>
> **For another models:** Please refer to our response to **Reviewer QhMC (Weakness 1)**
>
> **For more downstream tasks:** We extended LLaVA-v1.5 with two additional downstream tasks—**Remote Sensing (RS)** and **Medical (Med)** from MLLM-DCL—and compared it with **SPIDER** and **LoraSculpt**. The results are summarized in the table below:
>
> |Method|RS|OKVQA|OCRVQA|GQA|TextVQA|
> |---|---|---|---|---|---|
> |Zero-shot|32.27|57.99|66.2|61.93|58.23|
> |AWARe|79.68|52.52|59.6|56.93|53.47|
> |SPIDER|77.94|47.26|57.23|53.24|47.26|
> |LoraSculpt|78.27|48.32|55.45|53.78|49.56|
>
> |Method|Med|OKVQA|OCRVQA|GQA|TextVQA|
> |---|---|---|---|---|---|
> |Zero-shot|28.29|57.99|66.2|61.93|58.23|
> |AWARe|59.23|50.28|57.8|57.53|52.59|
> |SPIDER|57.75|46.54|54.79|51.79|46.79|
> |LoraSculpt|57.92|46.12|55.47|53.77|48.96|
>
> As shown in the results, AWARe demonstrates a significantly stronger ability to preserve upstream task capabilities.
>
> **We will open source our code for reproduce that result.**
>
> ### ${\color{#f26921}\text{Weakness-2:}}$
>
> **Response**: Please refer to our response to **Reviewer NcH2 (Question 2)**, where we provide a detailed discussion on this point.
>
> ### ${\color{#f26921}\text{Weakness-3:}}$
>
> **Response**: We appreciate the your feedback. While raw scores are intuitive, relative metrics ($\mathcal{R}$ and $\mathcal{E}$) were adopted to better capture the normalized ratios of knowledge acquisition and retention across different tasks. This ensures that the evaluation is not biased by the absolute magnitude of specific benchmarks.
>
> ### ${\color{#f26921}\text{Weakness-4:}}$
>
> **Response**:  Thank you for this important comment. We agree that Lemma 3.1 is not a proof that our saliency measure is uniquely optimal for preserving multimodal knowledge. Its role is narrower: under a direction-preservation objective, it shows why selecting neurons with the largest normalized directional contributions maximizes preserved directional energy. Our justification is therefore partly theoretical and primarily empirical. [Here we will directly quote our response to **Reviewer GKwX (Question 1)** on the relation between preserved energy and model generalization.] We will revise the paper to clarify the intended scope of the lemma and to better position it as one theoretical perspective on our design, complemented by the empirical evidence.
>
> ### ${\color{#f26921}\text{Weakness-5:}}$
>
> **Response**:  Please refer to our response to **Weakness 1**, where we provide a detailed discussion on this point.
>
> ### ${\color{#f26921}\text{Question-1:}}$
>
> > Can you evaluate AWARe on more base model families or at least one stronger alternative MLLM architecture beyond LLaVA-v1.5-7B?
>
> **Response**:  Please refer to our response to **Weakness 1**, where we provide a detailed discussion on this point.
>
> ### ${\color{#f26921}\text{Question-2:}}$
>
> > Can you report results on additional downstream tasks, especially tasks that are less similar to the current setup, to test whether the method generalizes beyond IconQA and COCO-Caption?
>
> **Response**:  Please refer to our response to **Weakness 1**, where we provide a detailed discussion on this point.
>
> ### ${\color{#f26921}\text{Question-3:}}$
>
> > How sensitive are the results to baseline hyperparameter tuning, especially for strong competitors like SPIDER, LoRASculpt, Orth-Reg, and Model Tailor?
>
> **Response**: We agree that hyperparameter tuning can affect absolute performance. To ensure a fair and conservative comparison, we used published results when available for strong baselines (notably LoRASculpt and SPIDER).
> Therefore, our results are not based on intentionally weak tuning of competitors. If anything, relying on their own reported strong results makes the comparison more conservative. While additional per-baseline sweeps might slightly change absolute numbers, we do not believe our main conclusion depends on under-tuned baselines: AWARe consistently shows a stronger retention--adaptation trade-off under this fair protocol.
>
> ### ${\color{#f26921}\text{Question-4:}}$
>
> > Have you tested whether AWARe still works well in a more sequential setting with multiple downstream fine-tuning stages, rather than a single downstream adaptation step?
>
> **Response**:  Please refer to our response to **Reviewer NcH2 (Question 2)**, where we provide a detailed discussion on this point.
>
> **Thank you for your thoughtful comments. We hope our response addresses your concerns. Please let us know if you have any further questions.**

---

> > ### Author Rebuttal · Reviewer_qF8R · 2026-04-02
> >
> > I thank the authors for the rebuttal and have updated the score accordingly.

---

> > > ### Author Response · Authors · 2026-04-03
> > >
> > > We sincerely thank you for your careful evaluation of our response and for your positive reassessment. We are glad that our rebuttal has addressed your main concerns. We will incorporate the clarifications discussed in the rebuttal into the revised paper to further improve its clarity and presentation.

---

### Official Review · Reviewer_QhMC · 2026-03-12

**Soundness:** 3
**Presentation:** 2
**Significance:** 2
**Originality:** 2
**Overall Recommendation:** 3
**Confidence:** 4

**Summary:**

The authors propose AWARe, a fine-tuning approach that alleviates catastrophic forgetting by dynamically controlling parameter updates according to activation patterns. It assigns activation-based importance scores to parameters, freezing those critical for retaining previous capabilities while permitting others to adapt to new tasks.

**Compliance With Llm Reviewing Policy:**

Affirmed.

**Final Justification:**

Thanks for the rebuttal. I keep my rating.

**Key Questions For Authors:**

See Weakness.

**Limitations:**

See Weakness.

**Strengths And Weaknesses:**

Strength:

1. The paper borrows the activation-aware saliency idea from AWQ and applies it to catastrophic forgetting, which represents a transfer of an existing technique to a different problem setting.

2. The method requires no additional modules and the profiling phase needs only a single forward pass over a few hundred samples, which reduces overhead compared to methods that require replay buffers or second-order statistics.

Weakness:

1. The entire evaluation uses one base model, LLaVA-v1.5-7B, with only two downstream tasks, IconQA and COCO-Caption. The paper needs results on more models and tasks.

2. The theoretical sections 3.5 and 3.6 are both labeled Theoretical Insight and largely repeat the same content. The lemma itself is a greedy knapsack argument and says nothing about why freezing these neurons prevents forgetting during gradient updates.

3. The retention ratio ρ is fixed at 30% with no method for choosing it when upstream validation data is unavailable.

4. Table 3 shows the harmonic mean stays nearly flat across retention ratios from 10% to 50%. This means the activation-based selection is not doing the work the paper claims, since similar results appear regardless of which neurons are frozen.

5. Comparisons with SPIDER and LoRASculpt reuse numbers from those original papers rather than re-running them under the same training setup, making the comparison unreliable.

---

> ### Author Rebuttal · Authors · 2026-03-31
>
> Thank you for acknowledging to our method. We provide explanations to your questions point-by-point in the following.
>
> ### ${\color{#f26921}\text{Weakness-1:}}$
>
> **Response**: We acknowledge the reviewer’s concern regarding the limited evaluation scope.
>
> **For more models:**
> To demonstrate generalizability, we conduct additional experiments on Qwen2.5-VL-7B trained with AWARe, evaluating on the Medical (Med) and Autonomous Driving (AD) subsets of [MLLM-DCL](https://github.com/bjzhb666/MLLM-CL). We analyse activation using MMMU datasets. The results are shown in the table below.
>
> |Method|Med|hallucion_bench|micro_vqa|docvqa|vstar_bench|ai2d|
> |---|---|---|---|---|---|---|
> |Origin|0.357|0.77|0.48|0.947|0.62|0.74|
> |SFT|0.521|0.33|0.07|0.599|0.01|0.04|
> |AWARe|0.477|0.76|0.48|0.913|0.65|0.73|
> |SPIDER|0.452|0.67|0.44|0.875|0.58|0.71|
> |LoraSculpt|0.465|0.72|0.46|0.863|0.56|0.69|
>
> |Method|AD|hallucion_bench|micro_vqa|docvqa|vstar_bench|ai2d|
> |---|---|---|---|---|---|---|
> |Origin|0.213|0.77|0.48|0.947|0.62|0.74|
> |SFT|0.560|0.5|0.39|0.19|0.24|0.37|
> |AWARe|0.463|0.76|0.44|0.901|0.6|0.71|
> |SPIDER|0.432|0.65|0.39|0.856|0.54|0.65|
> |LoraSculpt|0.445|0.69|0.42|0.847|0.52|0.69|
>
> **For more tasks:**
> We further evaluate LLaVA-v1.5 on MLLM-DCL under a continual instruction fine-tuning setting across five tasks, where performance remains strong (For further technical details, please refer to our response to **Reviewer NcH2 (Question 2)**.).
>
> **These results across different models and tasks further validate the effectiveness and generalizability of our framework. We will open source the related code.**
>
> Btw, I would like to mention that our method does not modify the model architecture or add new components. This indicates that our model can run directly on LLM inference engines (such as vLLM and SGLang), benefiting from their optimizations and high throughput.
>
> ### ${\color{#f26921}\text{Weakness-2:}}$
>
> **Response**: Thank you for this important comment. Sections 3.5 and 3.6 are indeed redundant; we will merge them.
>
> For any retained row we have $\Delta W_{k,:} = 0$. Thus, the “anchor” channels remain unchanged during downstream training, preventing destructive interference in the coordinates most important for upstream behavior.
>
> For more details on the **relation of preserved energy to model generalization**, please refer to our response to **Reviewer GKwX (Question 1)**.
>
> ### ${\color{#f26921}\text{Weakness-3:}}$
>
> **Response**: The choice of ρ is not closely tied to upstream tasks; we also computed activations on the MMMU dataset. We explained the rationale for selecting 30% in out response to **Reviewer NcH2 (Question 3)**.
>
> ### ${\color{#f26921}\text{Weakness-4:}}$
>
> **Response**: We thank the reviewer for the careful observation. We respectfully disagree that the relatively flat harmonic mean in Table 3 implies activation-based selection is ineffective. Table 3 studies the retention quota, while Table 2 isolates the selection criterion itself. Under the same 30% quota, activation-based selection outperforms random, weight-norm, and hybrid strategies, indicating that which neurons are frozen matters, not just how many.
>
> In addition, $H$ aggregates retention $R$ and learning efficiency $E$, and can mask a stability--plasticity trade-off. Under Global-Highest, $R$ increases from 95.8 to 99.1 as the quota rises from 10% to 50%, while $E$ peaks at 30% (108.4) and then drops to 107.3 at 50%. The flat $H$ therefore reflects compensation between improved retention and reduced plasticity.
>
> We use 30% by default because it is the smallest quota achieving the best $H$ ($103.2$), while remaining parameter-efficient and updating only about 17.5% of the parameters.
>
>
> ### ${\color{#f26921}\text{Weakness-5:}}$
>
> **Response**:Thank you for raising this concern. We reproduced LoRASculpt using its official code and reported hyperparameters (Rank = 32), but our results still deviated from those in the original paper, with some metrics differing by several points. To avoid reproduction-induced bias that could under-estimate the baseline, we therefore report the original LoRASculpt results in the main manuscript for fairness.
>
> ### IconQA
> |Methods|OKVQA|OCRVQA|GQA|TextVQA|Target|$\mathcal{R}$|$\mathcal{E}$|$\mathcal{H}$|
> |---|---|---|---|---|---|---|---|---|
> |LoRASculpt*|53.59|55.45|56.96|52.79|82.73|89.5|103.2|95.9|
> |LoRASculpt|53.52|59.50|57.63|53.76|85.26|91.8|106.4|98.6|
> |AWARe|56.95|65.10|60.83|57.66|86.92|98.4|108.4|103.2|
>
> ### COCO-Caption
> |Methods|OKVQA|OCRVQA|GQA|TextVQA|Target|$\mathcal{R}$|$\mathcal{E}$|$\mathcal{H}$|
> |---|---|---|---|---|---|---|---|---|
> |LoRASculpt*|46.74|60.20|53.66|49.45|124.35|86.0|130.1|103.5|
> |LoRASculpt|49.99|58.65|57.63|50.73|120.35|88.8|125.9|104.2|
> |AWARe|53.01|63.45|60.32|53.46|120.94|94.2|126.5|108.0|
>
> *: means reproduced by ourself.
>
> **Thank you for your thoughtful comments. We hope our response addresses your concerns. Please let us know if you have any further questions.**

---

> > ### Author Rebuttal · Reviewer_QhMC · 2026-04-03
> >
> > Thank you for the detailed response. My concerns have been addressed to some extent, and I hope the authors can incorporate this update into a future version of the paper.

---

### Official Review · Reviewer_NcH2 · 2026-03-12

**Soundness:** 3
**Presentation:** 3
**Significance:** 3
**Originality:** 3
**Overall Recommendation:** 3
**Confidence:** 4

**Summary:**

This paper investigates the challenge of catastrophic forgetting in Multimodal Large Language Models (MLLMs) during the fine-tuning phase. The authors introduce AWARe (Activation-Weighted Adaptive REtaining), a framework designed to identify and safeguard parameters critical to pre-trained knowledge using activation-based importance weights. By dynamically regularizing these essential parameters, AWARe seeks to strike a balance between task-specific adaptation and the preservation of foundational multimodal capabilities.

**Compliance With Llm Reviewing Policy:**

Affirmed.

**Final Justification:**

The rebuttal is well written and addresses a number of points from the initial review in a constructive manner. However, the clarifications provided are not sufficient to alter my overall judgment regarding the paper’s technical quality, originality, and significance. I therefore keep my original score.

**Key Questions For Authors:**

Please refer to the technical concerns and missing analyses detailed in the Weaknesses section, particularly the following:
* Can the authors provide a direct qualitative or quantitative comparison between activation-based and gradient-based importance maps?
* How does AWARe perform in a truly sequential multi-task learning setting (e.g., Task $A \to B \to C$)?
* Could the authors provide a sensitivity analysis for the selection ratio $\rho$ to demonstrate the method's stability?

**Limitations:**

yes

**Strengths And Weaknesses:**

Strengths
1. The adaptation of classical parameter-protection techniques to large-scale multimodal architectures via activation-weighted importance is a clever and well-motivated approach.
2. The manuscript provides a thorough empirical analysis, including comparisons with established LoRA-based and parameter-selection baselines such as LoRA, DoRA, and SPIDER.

Weaknesses
1. While the paper claims that "activation weights" are superior for identifying critical parameters, the main text lacks a direct, rigorous comparison (e.g., via a visualization or comparative table). Specifically, it remains unclear how activation-based importance maps differ from or outperform traditional gradient-based maps in the context of multimodal data.

2. The current experimental setup primarily evaluates single downstream tasks. In a standard "catastrophic forgetting" or "continual learning" paradigm, the model should ideally learn a sequence of tasks (e.g., Task $A \to B \to C$). The paper does not fully investigate whether the activation weights identified for Task $A$ remain valid or effective after Task $B$ is subsequently learned.

3. As noted in the Implementation Details (Page 14), AWARe relies on an "optimal selection ratio $\rho$." However, the manuscript lacks a formal sensitivity analysis (e.g., a line plot illustrating Performance vs. $\rho$). Without evidence of how the model performs when $\rho$ deviates, it is difficult to assess the robustness and practical utility of the method.

---

> ### Author Rebuttal · Authors · 2026-03-31
>
> Thank you for acknowledging to our method. We provide explanations to your questions point-by-point in the following.
>
> ### ${\color{#f26921}\text{Question-1:}}$
>
> > Can the authors provide a direct qualitative or quantitative comparison between activation-based and gradient-based importance maps?
>
> **Response**:
> Thank you for this helpful suggestion. We now provide a direct qualitative comparison between activation-based and gradient-based importance maps by visualizing the top 30% selected neurons for `q_proj`, `k_proj`, and `v_proj`. The two criteria exhibit clearly different selection patterns. The gradient-based importance map (in p2 norm to a row of a weight) is heavily concentrated on `v_proj`, with only sparse selections from `q_proj` and `k_proj`. In contrast, the activation-based map, although still dominated by `v_proj`, consistently identifies salient neurons in `q_proj` and `k_proj` as well. This suggests that gradient magnitude tends to focus on a narrower subset of parameters that are most sensitive to the current objective, whereas activation-based saliency captures a broader set of neurons that are consistently engaged by upstream representations across projection types. We believe this broader coverage helps explain the stronger retention–adaptation trade-off achieved by our method. We will incorporate this comparison and the corresponding visualization into the revised manuscript to make this distinction more explicit.
>
> **visualization for activation-based:** [Figure](https://picui.ogmua.cn/s1/2026/03/28/69c7acd10abe7.webp)
>
> **visualization for grad-based:** [Figure](https://picui.ogmua.cn/s1/2026/03/28/69c7acd0e2cad.webp)
>
> ### ${\color{#f26921}\text{Question-2:}}$
>
> > How does AWARe perform in a truly sequential multi-task learning setting (e.g., Task $A \to B \to C$)?
>
> **Response**: We evaluate AWARe, with MMMU to analyse activation, on the [MLLM-DCL](https://github.com/bjzhb666/MLLM-CL) [1] benchmark, a sequential instruction tuning benchmark comprising five tasks: RS (Remote Sensing), Med (Medical), AD (Autonomous Driving), Sci (Science), and Fin (Finance). The results demonstrate that AWARe maintains strong performance across the sequential learning pipeline. We compare our method with HiDe-LLaVA [2] and MoELoRA (detailed results are presented in the table) and find that our approach achieves superior performance. Additionally, after training on those five tasks, AWARe retains most of its upstream capabilities, scoring 53.61 on OKVQA, 65.25 on OCRVQA, 57.02 on GQA, and 55.31 on TextVQA which  demonstrating that AWARe retains the majority of its upstream capabilities. **We will open source the code to reproduce that result.**
>
> Detailed result are here:
>
> |AWARe|RS|Med|AD|Sci|Fin|
> |---|---|---|---|---|---|
> |RS|79.68|||||
> |Med|80.12|59.23||||
> |AD|80.38|59.23|53.57|||
> |Sci|79.31|54.71|52.34|52.53||
> |Fin|77.93|42.93|43.7|44.95|91.67|
>
> |HiDe-LLaVA|RS|Med|AD|Sci|Fin|
> |---|---|---|---|---|---|
> |RS|78.14|||||
> |Med|74.26|58.05||||
> |AD|74.90|42.94|39.65|||
> |Sci|75.43|44.91|38.33|46.44||
> |Fin|74.31|48.95|33.21|38.54|81.55|
>
> |MoELoRA|RS|Med|AD|Sci|Fin|
> |---|---|---|---|---|---|
> |RS|79.09|||||
> |Med|74.78|58.73||||
> |AD|77.69|43.72|51.47|||
> |Sci|76.87|43.79|32.81|48.67||
> |Fin|77.54|41.85|27.62|40.13|86.75|
>
> [1] Guo et al. "MCITlib: Multimodal Continual Instruction Tuning Library and Benchmark" arXiv 2025.
> [2] Guo et al. "Hide-llava: Hierarchical Decoupling for Continual Instruction Tuning of Multimodal Large Language Model" ACL 2025.
>
> ### ${\color{#f26921}\text{Question-3:}}$
>
> > Could the authors provide a sensitivity analysis for the selection ratio $\rho$ to demonstrate the method's stability?
>
> **Response**: We conducted extensive experiments on IconQA task with varying selection ratios $\rho$ = {1%, 10%, 30%, 40%, 50%, 70%, 90%}. Our results reveal a clear trade-off: freezing more parameters provides stronger protection for the upstream task, but the lack of parameter updates makes the downstream task harder to learn. We find that $\rho$ = 30% strikes an optimal balance between these competing objectives, representing a "sweet spot" where both upstream performance preservation and downstream task adaptation are achieved.
>
> |Ratio $\rho$|OKVQA|OCRVQA|GQA|TextVQA|Target|$\mathcal{R}$|$\mathcal{E}$|$\mathcal{H}$|
> |---|---|---|---|---|---|---|---|---|
> |1%|55.26|63.40|59.89|54.96|82.65|95.6|103.1|99.2|
> |10%|55.88|63.25|59.59|55.39|86.13|95.8|107.5|101.3|
> |30%|56.95|65.10|60.83|57.66|86.92|98.4|108.4|103.2|
> |40%|57.66|64.35|61.47|57.77|86.64|98.7|108.1|103.2|
> |50%|57.71|64.75|61.87|57.86|85.97|99.1|107.3|103.0|
> |70%|57.78|64.35|62.03|57.84|80.64|99.0|100.6|99.8|
> |90%|58.13|65.55|61.89|58.33|54.62|99.8|68.1|81.0|
>
> **Thank you for your thoughtful comments. We hope our response addresses your concerns. Please let us know if you have any further questions.**

---

> > ### Author Rebuttal · Reviewer_NcH2 · 2026-04-02
> >
> > Thank you for the detailed rebuttal and the additional experiments. I appreciate the clarifications, but they do not materially change my overall assessment, so I will maintain my original score. In particular, I remain concerned that the paper’s core setting is stabilization during specialization on a single downstream task, rather than standard sequential continual learning, and the broader catastrophic-forgetting/continual-learning framing still feels somewhat overstated.

---

> > > ### Author Response · Authors · 2026-04-02
> > >
> > > We thank the reviewer for raising this concern. While we agree that our primary setting is not identical to the canonical sequential continual learning setup, our experimental protocol follows the task settings established in **LoraSculpt** and **SPIDER**.
> > >
> > > **We would nevertheless like to emphasize that, in the continual-learning experiments added in response to multiple reviewers' requests, our method achieves strong and competitive performance across the evaluated settings. We therefore do not view the manuscript's framing as an overstatement**.
> > >
> > > Rather, these additional results provide concrete evidence that the proposed method remains effective beyond the original specialization setting, including in continual-learning scenarios for multimodal large models.
> > >
> > > **To further support the validity and transparency of our claims, we have included in the anonymous supplementary link the code for the multimodal large-model continual-learning experiments, together with the corresponding training curves and model checkpoints. We hope these materials help substantiate both the effectiveness of the method and the reproducibility of the reported results.**
> > >
> > > The model checkpoint, together with the corresponding training loss curve, has been made available at https://huggingface.co/anony574/llava-cl-Fin.
> > >
> > > We have also provided the anonymous repository at https://anonymous.4open.science/r/AWARe-206B, where the README.md includes detailed instructions for downloading the  MLLM-DCL dataset.
> > >
> > > ### Checkpoint evaluation script
> > > ```bash
> > > # Model path must corresponding to Current task
> > > # Current_Task is one of [RS Med AD Sci Fin]
> > > bash scripts/AWARe/eval_cl.sh <Model_Path> <Result_Dir> <Current_Task>
> > >
> > > # e.g.
> > > bash scripts/AWARe/eval_cl.sh ./outputs/llava-cl-Fin ./results/mllm-dcl Fin
> > > ```

---

### Decision · Program_Chairs · 2026-04-30

**Decision:**

Reject

**Comment:**

The reviewers acknowledge that mitigating catastrophic forgetting in Multimodal Large Language Models (MLLMs) is a timely and significant challenge. Reviewers find the method proposed in the paper, AWARe, to be simple, efficient (requiring only a single forward pass for calibration), and avoids additional inference overhead or architectural changes. However, significant concerns remain regarding the scope of evaluation, the theoretical depth, and the framing of the task as a true continual learning (CL) setup.

Strengths
* Methodological Efficiency: Reviewers (QhMC, qF8R) highlight that the activation-aware saliency approach is computationally light, requiring neither replay buffers nor second-order statistics.
* Empirical Performance: In the specific "one-step specialization" setting provided, the method shows competitive results against baselines like LoRA, DoRA, and SPIDER.
* Robust Ablations: Reviewer qF8R notes that the ablation studies—comparing activation-based selection against weight-based or random alternatives—are comprehensive and lend credibility to the method.

Weaknesses
* All reviewers (NcH2, QhMC, qF8R, GKwX) have pointed out that the paper suffers from narrow experimental setup, relying only on a single base model (LLaVA-v1.5-7B) and only two downstream tasks (IconQA, COCO-Caption). This limits the ability to claim state-of-the-art (SOTA) status or broad generalizability. Authors have tried to address this criticism through additional experiments during the rebuttal.
* Multiple reviewers (NcH2, qF8R) argue that the paper overstates its "continual learning" contributions. While the authors defend their protocol by citing LoRASculpt and SPIDER task settings, reviewers feel the primary experiments reflect one-step specialization (preserving pre-trained data while adding one task) rather than a canonical sequential CL paradigm (Tasks $A \rightarrow B \rightarrow C$).
* The theoretical sections (3.5 and 3.6) are criticized for being repetitive and providing a "greedy knapsack" argument that doesn't fundamentally explain why freezing specific neurons prevents gradient-based forgetting (QhMC, qF8R).
* Reviewers point out a lack of formal sensitivity analysis regarding the "optimal selection ratio" and question the novelty of the approach, viewing it as an adaptation of existing saliency ideas (like AWQ) to a new domain (GKwX).

Feedback to improve the quality of the paper:
1. Benchmarking Validity: Reviewer QhMC raised a concern that comparisons with SPIDER and LoRASculpt were reproduced from the original papers which can have a potentially distinct setup than the experimental setup in this paper, potentially making the comparisons unreliable.
2. Performance Flatness: There is a noted discrepancy where performance remains relatively flat across different retention ratios (10% to 50%), which may suggest the activation-based selection is not the primary driver of the observed results.

Overall, while the method is practical and the problem is relevant, the final version must frame the "specialization" vs. "continual learning" context clearly and demonstrate generalizability through relevantly additional empirical studies.